# Investigation on Ambipolar Current Suppression Using a Stacked Gate in an L-shaped Tunnel Field-Effect Transistor

**DOI:** 10.3390/mi10110753

**Published:** 2019-11-03

**Authors:** Junsu Yu, Sihyun Kim, Donghyun Ryu, Kitae Lee, Changha Kim, Jong-Ho Lee, Sangwan Kim, Byung-Gook Park

**Affiliations:** 1Inter-University Semiconductor Research Center (ISRC) and Department of Electrical and Computer Engineering, Seoul National University, Seoul 08826, Korea; liujs9860@gmail.com (J.Y.); si1226@snu.ac.kr (S.K.); show456852@snu.ac.kr (D.R.); syntax4me@gmail.com (K.L.); chterbox@snu.ac.kr (C.K.); jhl@snu.ac.kr (J.-H.L.); 2Department of Electrical and Computer Engineering, Ajou University, Suwon 16499, Korea

**Keywords:** L-shaped tunnel field-effect transistor, stacked gate, dual work function, ambipolar current

## Abstract

L-shaped tunnel field-effect transistor (TFET) provides higher on-current than a conventional TFET through band-to-band tunneling in the vertical direction of the channel. However, L-shaped TFET is disadvantageous for low-power applications because of increased off-current due to the large ambipolar current. In this paper, a stacked gate L-shaped TFET is proposed for suppression of ambipolar current. Stacked gates can be easily implemented using the structural features of L-shaped TFET, and on- and off-current can be controlled separately by using different gates located near the source and the drain, respectively. As a result, the suppression of ambipolarity is observed with respect to work function difference between two gates by simulation of the band-to-band tunneling generation. Furthermore, the proposed device suppresses ambipolar current better than existing ambipolar current suppression methods. In particular, low drain resistance is achieved as there is no need to reduce drain doping, which leads to a 7% enhanced on-current. Finally, we present the fabrication method for a stacked gate L-shaped TFET.

## 1. Introduction

A tunnel field-effect transistor (TFET) has attracted attention as a candidate for low-power applications because of its low subthreshold swing and low off-current compared with the metal-oxide-semiconductor field-effect transistor (MOSFET) [1,2,3,4,5]. Since a working principle of TFET relies on band-to-band tunneling (BTBT), TFET can achieve under 60 mV/decade subthreshold swing which acts as a limit on MOSFET [5,6,7]. However, TFET has a limitation in its on-current, which is lower than that of the conventional MOSFET because of low BTBT rates [8]. To solve this problem, an L-shaped TFET using vertical BTBT has been proposed [9]. Nevertheless, it has a disadvantage of ensuing large ambipolar current due to the tunneling layer deposited on the gate-drain overlap region during the selective epitaxial-layer growth (SEG) process [10]. Since ambipolar current contributes to the increase of the off-current, finding a method to reduce it is an important issue. Reduced drain doping and gate-drain underlap have been suggested as strategies for eliminating ambipolar current [11,12,13,14]. However, the method reducing drain doping concentration has drawbacks in terms of decreased on-current because of the increased drain resistance and in terms of increased Miller capacitance due to increased gate-drain coupling, which leads to the degradation of resistor–capacitor (RC) switching characteristics [12]. The gate-drain underlap also has a drawback which limits the scalability.

Therefore, in this paper, we propose a method of suppressing ambipolar current by simply stacking the gates utilizing the structural features of L-shaped TFET. First, the structure of the proposed device and simulation method are described. Next, the electrical characteristics of the device are analyzed, which is followed by comparisons to other methods of suppressing ambipolar current. Finally, the fabrication method is presented for the stacked gate L-shaped TFET.

## 2. Device Structures and Simulation Methods

Figure 1a–d show the schematic designs of the single gate, stacked gate L-shaped TFET and the other devices with gate-drain underlap applied to each of the two devices. All devices are based on silicon and share the same doping concentration except low drain doping device (1 × 10^19^ cm^−3^ on drain). Abrupt doping profile can be formed because of in-situ doping during epitaxy, especially at the source [15]. Work function of the top gate (*ϕ*_G2_) is fixed at 4.5 eV and its height (*H*_G2_) is 88 nm. The bottom gate work function (*ϕ*_G1_) varies from 4.0 to 4.5 eV and its height (*H*_G1_) is 10 nm. The source height is adjusted to 65 nm, which allows the SEG tunneling layer between the source and the gate to be controlled by the top gate while the bottom channel is controlled by the bottom gate. The vertical tunneling thickness (*L*_t_) is 4 nm and the underlap length (*L*_un_) is 9 nm. All design parameters are summarized in Table 1. In order to verify the suppression of ambipolar current due to the stacked gates structure, electrical characteristics of each device are investigated through Synopsys Sentaurus™ Technology Computer-Aided Design (TCAD) two-dimensional (2D) device simulation. The nonlocal BTBT model is applied for investigation of ambipolar current in L-shaped TFET since this model takes tunneling effect into consideration based on energy band profile. Two tunneling model coefficients *A*_Si_ = 4.0 × 10^14^ cm^−1^s^−1^, *B*_Si_ = 9.9 × 10^6^ V/cm, *A*_SiGe_ = 3.1 × 10^16^ cm^−1^s^−1^ and *B*_SiGe_ = 7.1 × 10^5^ V/cm from [16] are used in this work.

## 3. Results

### 3.1. Ambipolar Suppression of Stacked Gate L-Shaped TFET

As shown in Figure 2a, the ambipolar current is significantly decreased in stacked gate L-shaped TFET because *ϕ*_G1_ is lower than *ϕ*_G2_, which leads to larger channel potential at the drain side. Meanwhile, on-current remains constant because *ϕ*_G2_ is the same as that of the single gate L-shaped TFET so that the same amount of electrostatic potential is applied to the SEG tunneling layer. Consequently, the on-state region remains unchanged while the off-state region expands [(ii) to (iv)] and the ambipolar state region contracts [(i) to (iii)]. The on-state region, off-state region and ambipolar state region are defined with the constant current method. Figure 2b shows that the tunneling barrier width between the channel and the drain becomes thicker in the stacked gate L-shaped TFET due to the stronger potential applied to the channel. As a result, the BTBT rate of stacked gate L-shaped TFET significantly decreases in the ambipolar state (Figure 3). In addition, considering the relationship of the potential applied to the channel according to the work function of the gate, the ambipolar state region in the transfer curve will be shifted to the left by decreasing *ϕ*_G1_, which will be covered in a later subsection.

As illustrated in Figure 4, the ambipolar current of stacked gate L-shaped TFET with underlap is the most suppressed. Non-stacked devices have similar off-state region sizes, while stacked devices have an expanded off-state region and reduced ambipolar state region. Comparing with the single gate L-shaped TFET (without underlap), the stacked gate L-shaped TFET (without underlap) shows ambipolar current (drain current at *V*_GS_ = −1 V) and ambipolar region to be reduced and contracted by 12 times and by 0.5 V, respectively. This advantage further reduces off-current and makes it less sensitive to process variations.

### 3.2. Gate1 Work Function (ϕ_G1_) Split

Figure 5 shows the transfer curves of stacked gate L-shaped TFET with various *ϕ*_G1_. As the *ϕ*_G1_ decreases, the ambipolar state region contracts and the off-state region expands because the energy band of the channel drops downward (Figure 6). It leads to thickening of the tunneling barrier width between channel and drain, reducing BTBT rates, as shown in Figure 7.

### 3.3. Resistance/on-Current

Increase in on-current is beneficial in terms of RC characteristic due to a reduction in the resistance. Figure 8a displays the resistance network in stacked gate L-shaped TFET when the device is in the on-state. Considering that the BTBT generation that contributes to the on-current occurs in two places near the source, the resistance network can be described as above. Since there is no need to lower the drain doping to suppress the ambipolar current, the drain resistance does not increase and it leads to higher on-current than the conventional method. Moreover, increasing the BTBT rate, for example by changing the source from Si to SiGe, reduces the tunneling resistance (*R*_TUN1_, *R*_TUN2_) and makes the effect of drain resistance more critical. Figure 8b exhibits the on-current with the drain doping concentration in stacked gate L-shaped TFET using SiGe on the source. As a result, up to 7% of an on-current gain can be achieved with an L-shaped TFET.

### 3.4. Process Flow

The stacked gate L-shaped TFET can be easily fabricated by the repetition of deposition and etch-back processes, unlike the planar structure in which the lithography process is necessary to form the stacked gate. Figure 9 illustrates the key process steps for stacked gate L-shaped TFET. The other processes before the stacked gate are described in [15]. After the gate dielectric deposition, Gate 2 atomic layer deposition (ALD) process (Figure 9a) is followed by chemical mechanical polishing (CMP) (Figure 9b). Then the etch-back process is done to recess Gate2 under the source (Figure 9c). Finally, Gate1 is repeatedly deposited and CMP is done (Figure 9d). The process flow for the gate stack is explained in [17].

## 4. Summary

In this study, we have successfully suppressed the ambipolar current of L-shaped TFET. The results prove that the ambipolar current can be efficiently suppressed by stacking the gates and using a low *ϕ*_G2_. Compared with the other strategies for suppressing ambipolar behavior, the stacked gate method shows the best performance in terms of ambipolar current, on-current and self-aligned process feasibility. Consequently, the proposed device will be a better candidate for the future generation of ultra-low-power circuits.

## Figures and Tables

**Figure 1 micromachines-10-00753-f001:**
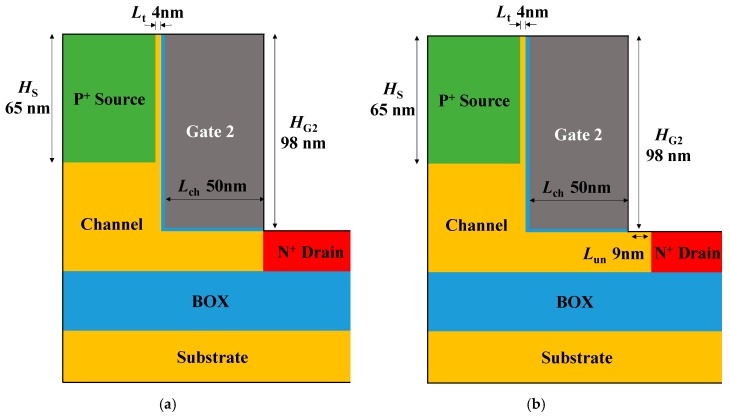
Schematic designs of (**a**) (conventional) single gate L-shaped tunnel field-effect transistor (TFET), (**b**) single gate L-shaped TFET with underlap, (**c**) stacked gate L-shaped TFET and (**d**) stacked gate L-shaped TFET with underlap. In order to compare the suppression of the ambipolar current for each method, the simulations were conducted according to the schematics above.

**Figure 2 micromachines-10-00753-f002:**
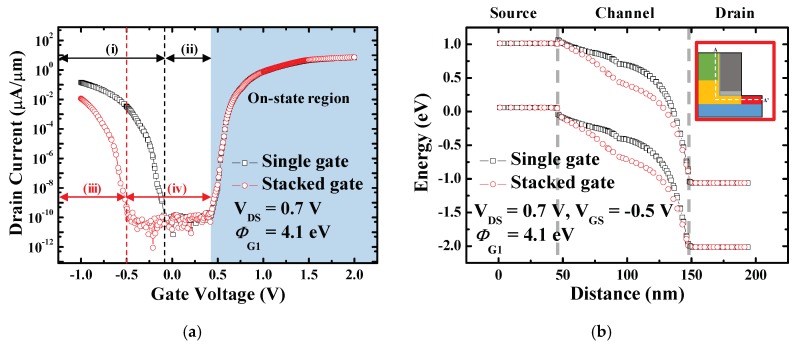
(**a**) Transfer curves of single gate L-shaped TFET and stacked gate L-shaped TFET with 0.7 V-drain voltage (*V*_DS_). Off-state and ambipolar state region of each device is distinguished by (i)–(iv). Ambipolar state region of the single gate (i), the off-state region of the single gate (ii), ambipolar state region of the stacked gate (iii) and the off-state region of the stacked gate (iv). It is shown that the ambipolar current is suppressed and the ambipolar state region contracts in the stacked gate L-shaped TFET; (**b**) Energy band diagram at gate voltage (*V*_GS_) = −0.5 V for single gate L-shaped TFET and stacked gate L-shaped TFET. They are obtained from source-to-drain along the cutline which is indicated in the inset (A-A’). It is presented that the tunneling barrier between channel-to-drain becomes thicker in the stacked gate L-shaped TFET.

**Figure 3 micromachines-10-00753-f003:**
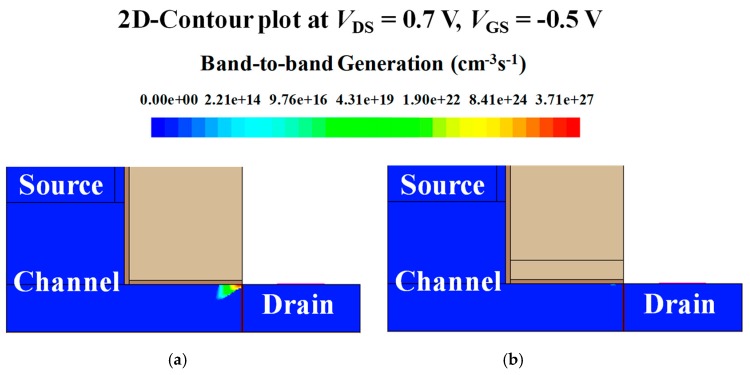
2-D contour plot of band-to-band tunneling (BTBT) generation for (**a**) single gate and (**b**) stacked gate L-shaped TFET. Ambipolar current suppression is observed from the decreased BTBT rates in the stacked gate L-shaped TFET.

**Figure 4 micromachines-10-00753-f004:**
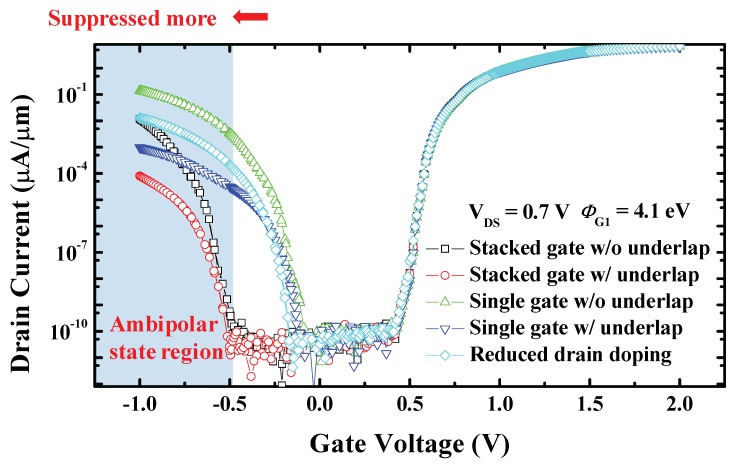
Transfer characteristics for stacked gate L-shaped TFET, stacked gate L-shaped TFET with underlap, low drain doping L-shaped TFET, (conventional) single gate L-shaped TFET, and single gate L-shaped TFET with underlap at *V*_DS_ = 0.7 V. It is illustrated that the ambipolar state region significantly contracts in the stacked gate L-shaped TFET.

**Figure 5 micromachines-10-00753-f005:**
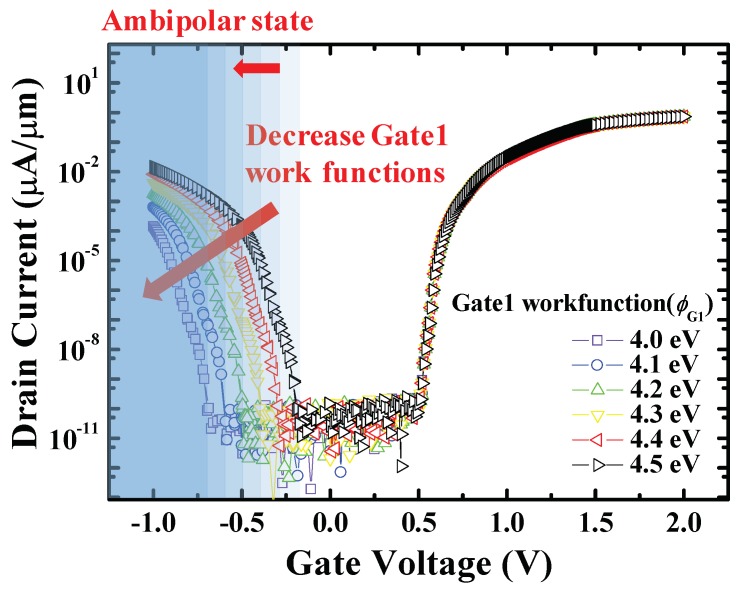
Transfer curves of stacked gate L-shaped TFET with various *ϕ*_G1_ (at *V*_DS_ = 0.7 V). As *ϕ*_G1_ decreases, ambipolar current is suppressed more and also ambipolar state region contracts. It can be interpreted as if only the left part of the transfer curve (*V*_GS_ < 0) is shifted to the left.

**Figure 6 micromachines-10-00753-f006:**
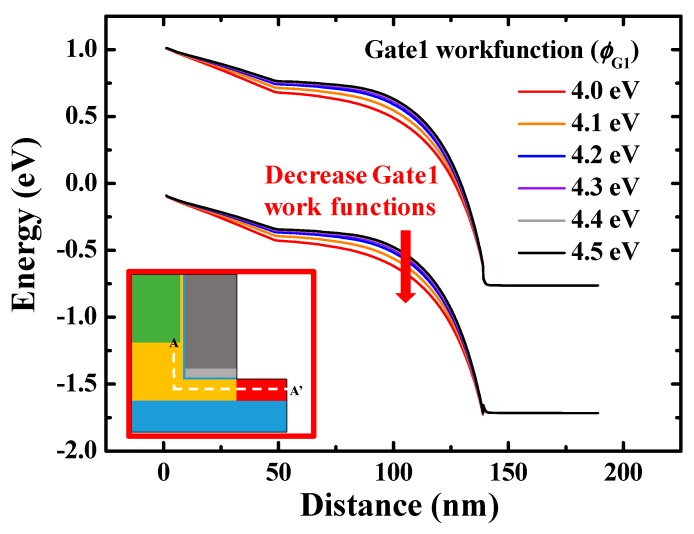
Energy band diagrams with various *ϕ*_G1_ (at *V*_DS_ = 0.7 V). They are obtained from channel-to-drain along the cutline which is indicated in the inset (A-A’). As *ϕ*_G1_ decreases, the tunneling barrier between the channel and the drain becomes thicker.

**Figure 7 micromachines-10-00753-f007:**
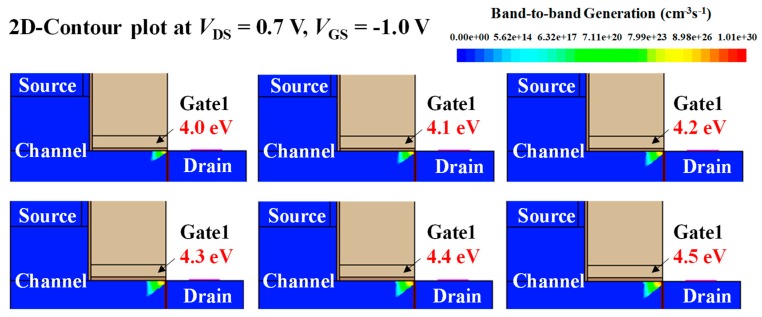
2-D contour plot of BTBT generation for various *ϕ*_G1_. As *ϕ*_G1_ decreases, BTBT rate decreases.

**Figure 8 micromachines-10-00753-f008:**
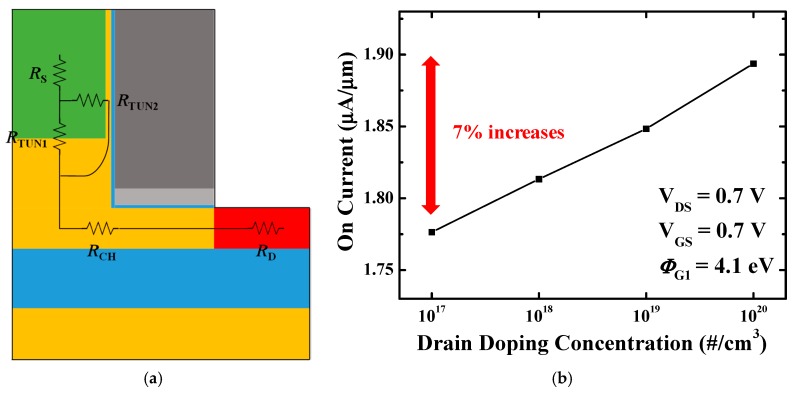
(**a**) Resistance network when the device is in the on-state. *R*_TUN1_ and *R*_TUN2_ are the tunneling resistances; (**b**) on-current with the drain doping concentration. After changed the source material to SiGe, 7% on-current increases with the drain doping concentration.

**Figure 9 micromachines-10-00753-f009:**
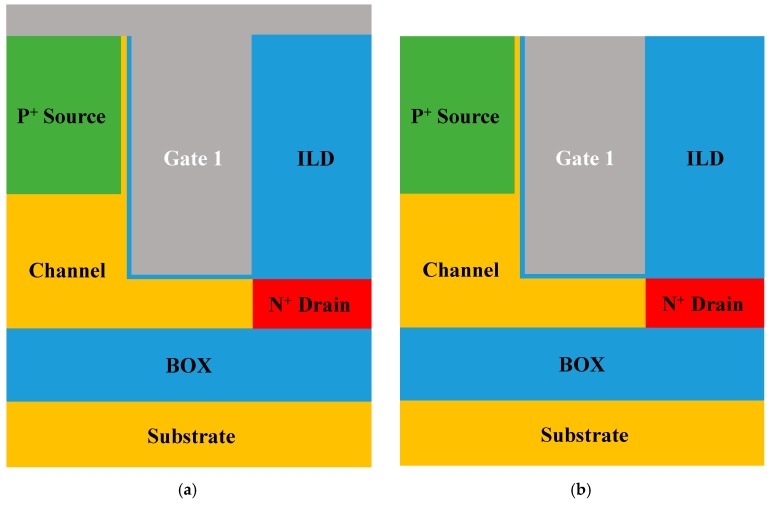
Key process steps for stacked gate L-shaped TFET. (**a**) Gate 2 is deposited by atomic layer deposition (ALD); (**b**) Gate 2 chemical mechanical polishing (CMP); (**c**) etch-back process; (**d**) Gate1 ALD & CMP.

**Table 1 micromachines-10-00753-t001:** Simulation parameters used for this work.

Parameters	Definitions	Value
*N* _S_	Source doping concentration	Boron, 1 × 10^20^ cm^−3^
*N* _B_	Channel doping concentration	Boron, 1 × 10^16^ cm^−3^
*N* _D_	Drain doping concentration	Arsenic, 1 × 10^19^ cm^−3^, 1 × 10^20^ cm^−3^
*H* _S_	Source height	65 nm
*H* _G1_	Gate1 height	88 nm
*H* _G2_	Gate2 height	10 nm
*L* _t_	Vertical tunneling thickness	4 nm
*L* _ch_	Lateral channel length	50 nm
*L* _un_	Gate-drain underlap length	9 nm
*T* _B_	Body thickness	20 nm
*T* _OX_	Gate oxide thickness	2 nm
*V* _DS_	Drain voltage	0.7 V
*ϕ* _G1_	Gate1 work function	4.0–4.5 eV
*ϕ* _G2_	Gate2 work function	4.5 eV

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
