# Peer review of "Investigation on Ambipolar Current Suppression Using a Stacked Gate in an L-shaped Tunnel Field-Effect Transistor"

_micromachines, 2019, doi:10.3390/mi10110753_

Round 1

Reviewer 1 Report

Please include the thickness of each layer in the TFT in Fig 1. Add more details in the fig captions.

How is the data in Fig 2b obtained? Experimental or simulation?

Reduce font size in Fig 3. Add more detail in caption.

Don't let data overlap with the legends in Fig 4 and 5.

Add more detail in the captions of Fig 5, 6, 7, 8. Each Fig and caption should be self-explanatory. They should tell the reader a summarized description of the figure. 

How thick are the layers in Fig 9?

Have you made the transistors experimentally and tested it? It will be good to have the simulation verified with experimental data/results.

Reviewer 2 Report

In this paper, the authors introduced a stacked gate to L-shaped TFET in order to suppress the off-current. The idea of using two gate metal with different metal work function is interesting and reasonable for the purpose. And The device results shown in the manuscript are promising. On the other hand, I would be more interested to see more details on the fabrication process which is more important to the reproducibility. SEM or AFM images would be very helpful to check the device structure especially for those special steps after fabricating the first metal gate. Meanwhile, I'm curious about the surface status for the first gate after CVD/CMP since it is close to 10nm. Is it still a continues layer?

Also does the thickness of Gate1 affect the overall performance. In this study, fixed dimensions are presented. Did the authors did series studies on varying those dimension parameters? 

Round 2

Reviewer 1 Report

Thanks for the edits. It is better. Please try to explain more and detailed for the figure captions.

Author Response

Thank you for your kind comment. We have added a more detailed description in the figure captions so that the figure can be self-explanatory.

Corresponding change in manuscript: Yes

Location of Change: Figure 2, 4

Reviewer 2 Report

The replies from the authors are solid. Most questions are addressed. Since this is an experimental focused study, I think direct SEM/AFM images are necessary.

Author Response

Thank you for your kind comment. We agree with you that SEM images are necessary in experimental focused study. But the focus of this study is not on experiments but on the investigation of the novel method by TCAD simulation. Nevertheless, SEM images of the previous study (single gate L-shaped TFET) can be found in [10] and the SEM images of the stacked gate can be found in [17].

We are sorry that we cannot provide direct SEM images of our proposed device, but we hope you give credit for our proposing the new method for suppressing ambipolar current.